# On a Relation Between the Rate-Distortion Function and Optimal Transport

**Eric Lei, Hamed Hassani and Shirin Saeedi Bidokhti**
Dept. of Electrical and Systems Engineering, University of Pennsylvania, USA
{elei,hassani,saeedi}@seas.upenn.edu

## ABSTRACT

We discuss a relationship between rate-distortion and optimal transport (OT) theory, even though they seem to be unrelated at first glance. In particular, we show that a function defined via an extremal entropic OT distance is equivalent to the rate-distortion function. We numerically verify this result as well as previous results that connect the Monge and Kantorovich problems to optimal scalar quantization. Thus, we unify solving scalar quantization and rate-distortion functions in an alternative fashion by using their respective optimal transport solvers.

**Rate-Distortion.** Let $X \sim P_X$ be the source supported on $\mathcal{X}$. Let $\mathcal{Y}$ be the reproduction space, and $\rho : \mathcal{X} \times \mathcal{Y} \to \mathbb{R}_{\geq 0}$ be a distortion measure. The asymptotic limit on the minimum number of bits required to represent $X$ with average distortion at most $D$ is given by the rate-distortion function Cover & Thomas (2006), defined as

$$R(D) := \inf_{P_{Y|X}: \mathbb{E}_{P_{X,Y}}[\rho(X,Y)] \leq D} I(X;Y). \tag{1}$$

Any rate-distortion pair $(R, D)$ satisfying $R > R(D)$ is achievable by some lossy source code, and no code can achieve a rate-distortion less than $R(D)$.

$R(D)$ has the following alternate form (Cover & Thomas, 2006, Ch. 10),

$$R(D) = \inf_{Q_Y} \inf_{P_{Y|X}: \mathbb{E}_{P_{X,Y}}[\rho(X,Y)] \leq D} D_{\mathsf{KL}}(P_{X,Y} || P_X \otimes Q_Y). \tag{2}$$

Due to the convex and strictly decreasing properties Cover & Thomas (2006) of $R(D)$, it suffices to fix $\lambda > 0$ and solve

$$\inf_{Q_Y} \inf_{P_{Y|X}} D_{\mathsf{KL}}(P_{X,Y} || P_X \otimes Q_Y) + \lambda \mathbb{E}_{P_{X,Y}}[\rho(X,Y)]. \tag{3}$$

A solution to (3) corresponds to a point on $R(D)$ corresponding to $\lambda$. The Blahut-Arimoto (BA) algorithm Blahut (1972); Arimoto (1972) solves (2) by alternating steps on $P_{Y|X}$ and $Q_Y$ until convergence. Sweeping over $\lambda$ gives the entire rate-distortion curve.

**Optimal Transport.** We consider optimal transport (OT) under the Kantorovich formulation, which finds the minimum distortion coupling $\pi$ between measures $\mu$ and $\nu$[1],

$$W(\mu, \nu) := \inf_{\pi \in \Pi(\mu,\nu)} \mathbb{E}_{X,Y \sim \pi}[\rho(X,Y)]. \tag{4}$$

Under certain conditions, the optimal coupling is induced by a fixed mapping, known as the Monge map. The Kantorovich problem is often regularized with an entropy term,

$$S_\epsilon(\mu, \nu) := \inf_{\pi \in \Pi(\mu,\nu)} \mathbb{E}_\pi[\rho(X,Y)] + \epsilon D_{\mathsf{KL}}(\pi || \mu \otimes \nu), \tag{5}$$

which is known as entropy-regularized optimal transport, with $\epsilon > 0$. For discrete measures $\mu, \nu$, (5) can be solved efficiently using the Sinkhorn algorithm Knopp & Sinkhorn (1967); Sinkhorn (1964).

**Related Work.** A connection between source coding and optimal transport was made in a talk given by Gray (2013), who discusses how scalar quantizers can be found through an extremal

---

[1]A joint distribution that marginalizes to $\mu$ and $\nu$.

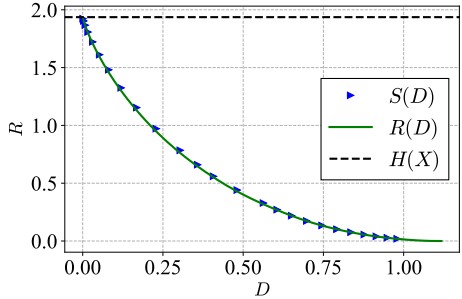
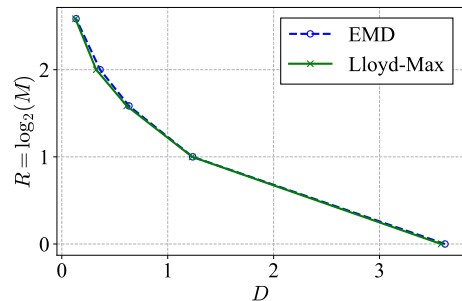

Figure 1: Equivalence of $S(D)$ and $R(D)$ on a 5-atom discrete source with $\rho(x, y) = (x - y)^2$.

Figure 2: Equivalence of extremal EMD and Lloyd-Max for $M$-level scalar quantization.

Monge/Kantorovich problem, and alludes to a similar connection for Shannon's rate-distortion function. Here, we concretely provide $R(D)$'s connection with entropic OT and discuss how their respective computational methods (Blahut-Arimoto and Sinkhorn-Knopp) can compute $R(D)$. In a similar vein, we empirically verify Gray (2013)'s results and show that Lloyd-Max and Earth Mover's distance can both compute optimal scalar quantizers. A similar result relating rate-distortion with entropic OT was also reported in Wu et al. (2022) which was unbeknownst to us at the time.

**Main Result.** We first show that entropic OT can be used to upper bound $R(D)$. First, observe that the inner minimization problem in (3) looks similar to the entropic OT problem. Let us define

$$S(D) := \inf_{Q_Y} \inf_{\substack{\pi \in \Pi(P_X, Q_Y): \\ \mathbb{E}_\pi[\rho(X,Y)] \leq D}} D_{\mathsf{KL}}(\pi \| P_X \otimes Q_Y), \tag{6}$$

which we call the *Sinkhorn-distortion function*, and is an extremal entropic OT distance w.r.t. $P_X$. Similar to $R(D)$, we can trace out $S(D)$ by sweeping over $\lambda > 0$, and solving the inner minimization (5), and then optimizing over all $Q_Y$, which is a convex problem in $Q_Y$ Feydy et al. (2019). It is clear that $R(D) \leq S(D)$ by comparing (6) and (2). Next, we show that without further assumptions, $R(D)$ and $S(D)$ are equivalent.

**Theorem 1.** *For any source $P_X$ and distortion function $\rho : \mathcal{X} \times \mathcal{Y} \to \mathbb{R}_{\geq 0}$, it holds that*

$$R(D) = S(D). \tag{7}$$

See Sec. A.1 for the proof. We numerically verify the equivalence in Fig. 1 on a discrete source with 5 atoms under squared-error distortion. For $R(D)$, we use Blahut-Arimoto, and for $S(D)$, we solve the convex problem using SQP solvers Kraft (1988) with $Q \mapsto S_\epsilon(P_X, Q)$ as the objective function, showing that the two different objectives result in the same function.

**Discussion.** Observe that the joint $P_{X,Y} = P_X P_{Y|X}$ defined in (2) marginalizes to $P_X$ but not necessarily $Q_Y$, whereas the coupling $\pi$ in (6) marginalizes to both. This result says that the additional $Q_Y$ marginalization constraint in $S(D)$ plays no role when both objectives are infimized over $Q_Y$. In computing $R(D)$, this provides an alternative to Blahut-Arimoto: solve (6) directly over $Q_Y$, using Sinkhorn iterations as a subroutine when evaluating the objective function (or its gradient). A symmetrized variant of the Sinkhorn-distortion function is often used to solve generative modeling tasks with Sinkhorn divergences Genevay et al. (2018); Salimans et al. (2018); Shen et al. (2020), where one wishes to find some $Q_Y \approx P_X$ by solving $\min_{Q_Y} S_\epsilon(P_X, Q_Y)$. However, if one leaves the objective un-symmetrized, the optimal $Q_Y^*$ and coupling $\pi^*$ are actually $R(D)$-achieving distributions with $\lambda = 1/\epsilon$.

We also verify that in discrete settings, the extremal non-entropic OT function $\min_{Q_Y:|Q_Y|\leq M} W(P_X, Q_Y)$, where $|Q_Y|$ is the size of $Q_Y$'s alphabet, is equivalent to optimal scalar quantization of $P_X$ as shown in Gray (2013). In Fig. 2, we solve the $\min_{Q_Y:|Q_Y|\leq M} W(P_X, Q_Y)$ on a 10-atom source using a linear program to compute the Earth Mover's distance (EMD) $W(\cdot, \cdot)$ and pass the function to a SQP solver as before. The achieved rate-distortion is equivalent to that of Lloyd-Max ($M$-means).

URM Statement

All authors meet the URM criteria of ICLR 2023 Tiny Papers Track.

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

## A  Appendix

### A.1  Proofs

**Theorem 1.** *For any source $P_X$ and distortion function $\rho : \mathcal{X} \times \mathcal{Y} \to \mathbb{R}_{\geq 0}$, it holds that*

$$R(D) = S(D). \tag{7}$$

*Proof.* From (Cover & Thomas, 2006, Ch. 9), the optimizers $Q_Y^*, P_{Y|X}^*$ of (3) for a fixed $\lambda > 0$ satisfy

$$\frac{dP_{Y|X=x}^*}{dQ_Y}(x,y) = \frac{e^{-\lambda \rho(x,y)}}{\int_{\mathcal{Y}} e^{-\lambda \rho(x,\tilde{y})} dQ_Y^*}, \tag{8}$$

$$Q_Y^* = \int_{\mathcal{X}} dP_{Y|X}^* dP_X, \tag{9}$$

simultaneously, which achieves a unique point on $R(D)$ corresponding to $\lambda$. To show that $S(D)$ achieves the same objective as $R(D)$ on the same $P_X$ and distortion measure, it suffices to show that the $R(D)$-optimal $Q_Y^*$ and $P_{Y|X}^*$ are feasible for $S(D)$, since $R(D) \leq S(D)$. From (Peyré & Cuturi, 2019, Ch. 4, Prop. 4.3), the optimal coupling $\pi^*$ in entropic OT is unique and has the form

$$\frac{d\pi^*}{dP_X dQ_Y}(x,y) = u(x)e^{-\lambda \rho(x,y)}v(y), \tag{10}$$

where $u(x), v(y)$ are dual variables that ensure $\pi^*$ is a valid coupling. The $R(D)$-optimal joint distribution $P_X P_{Y|X}^*$, which is guaranteed to be a coupling between $P_X$ and $Q_Y^*$ due to (9), indeed has the form

$$\frac{dP_X P_{Y|X}^*}{dP_X dQ_Y^*}(x,y) = \frac{1}{\int_{\mathcal{Y}} e^{-\lambda \rho(x,y')} dQ_Y^*} \cdot e^{-\lambda \rho(x,y)} \cdot 1, \tag{11}$$

where the first term only depends on $x$ and the last term only depends on $y$. Since $R(D)$ is a lower bound of $S(D)$, we are done. $\square$

