# OpenReview forum: "On a Relation Between the Rate-Distortion Function and Optimal Transport"
_ICLR.cc/2023/TinyPapers — Submitted to Tiny Papers @ ICLR 2023_

### Official Review · Reviewer_oeQU · 2023-03-22

**Confidence:** 3

**Summary Of Contributions:**

This paper establishes a theoretical connection between rate-distortion theory and optimal transport theory, which is then empirically validated. This connection allows for the solving of rate-distortion functions through their respective optimal transport solvers rather than directly.

**Rating:**

Great Start (GS): a submission which meets some of the reviewing criteria but has room for improvement

**Strengths And Weaknesses:**

Strengths:

- Clarity: Relevant literature for the connection between optimal transport theory and rate-distortion functions is thoroughly discussed, including relevant talks. Proofs and subsequent empirical evaluations are clear and communicated effectively.

- Correctness: Proofs appear to be correct and are further empirically validated to ensure correctness.

- Reproducibility: Proofs are well laid out, symbols are defined, everything can be followed/reproduced

- Follows requirements: Is within the page limit, adheres to the code of conduct

Weaknesses:

- Clarity: Relevant literature to the connection between rate-distortion and optimal transport theory is well discussed, but background and definitions for each of these individually could be more present. Additionally, the relationship between the two was well proven in this work but discussions on implications of this new relationship also could have been more present to highlight the importance of this relationship.

**Suggested Changes:**

Thank you for your submission! I think the proofs are well laid out and provide a great, solid foundation for this work. I also appreciated that there were empirical validations of the relationship found. Overall, this work met many of the criteria but has some room for improvement in clarity of the problem and implication of the findings.

First, I think that the relationship between optimal transport theory and rate-distortion functions was discussed well. However, I think that  providing some more details on each of this individual topics, such as definitions, uses, etc. (even just a couple sentences for each one) would drastically improve the clarity of the paper and set up relevant background/preliminary knowledge, especially for readers who may not be familiar with either of these two topics.

Second, this work does a great job at proving and validating this relationship between optimal transport and rate-distortion functions. However, it would be great to add some discussion on why this relationship matters and what the implications of these findings are, particularly since these two areas do seem unrelated (as is mentioned in the abstract). Adding these details would greatly increase the clarity of the paper with respect to the importance of the findings and any potential future relevant uses for this relationship.

---

### Official Review · Reviewer_gjgu · 2023-03-27

**Confidence:** 4

**Summary Of Contributions:**

This paper explores the relation between rate-distortion from information theory and Optimal Transport. They empirically and theoretically show that extremal entropic OT distance is directly related to the rate-distortion function.

**Rating:**

High Potential (HP): a submission which meets the reviewing criteria and has potential to make an impact on the field

**Strengths And Weaknesses:**

### Strengths :
-  I think this work is of very high quality, very well written and with very consistent and easy to follow notation. It was certainly a fun read.
-  The brief introduction to the topic is certainly nice to have as the reader can get acquainted to the notation used throughout the paper.
-  This paper not only show that extremal entropic OT distance and rate distortion function are equivalent but also shows some nice empirical results.


### Weaknesses :
-  I do believe this paper could be structured a little better with section headings instead of topic names.

Although I do not think this work makes significant discoveries theoretically but they do provide adequate verification of the ongoing discussion in OT.

**Suggested Changes:**

-  I would recommend the authors to restructure the paper according to well-defined headings instead of paragraphs.

---

### Author Response · Authors · 2023-05-31
**Opt-in for archival**

We confirm that we would like to opt in for archival.

---

### Comment · Area_Chair_UrBe · 2023-06-06
**Check for Archival**

This work meets the threshold for archival, contents the URM statement and is deanonymized

---

### Meta-Review · Area_Chair_UrBe · 2023-04-08

**Recommendation:** Invite to present
**Confidence:** 3

**Metareview:**

This paper explores the connection between rate-distortion and optimal transport from both theoretical and empirical perspectives. The authors state that extremal entropic OT distance is directly related to the rate-distortion function, allowing for the solving of rate-distortion functions through their respective optimal transport solvers rather than directly.

Both reviewers acknowledge the CCR and contribution of this paper. This paper could be improved further after revision. Based on the reviewers' comments, the AC suggests:
* a better organization (e.g., section headings instead of topic names);

* add preliminaries, such as background and definitions of rate-distortion and optimal transport theory, for readers that are not familiar with the topics;

* add more discussions on the implications of the findings.

Overall, based on the review criteria of the ICLR TinyPaper Track, it meets the CCR standard. Please carefully revise and proofread the paper following both reviewers' comments.


**Summary:**

This paper explores the connection between rate-distortion and optimal transport from both theoretical and empirical perspectives. The authors state that extremal entropic OT distance is directly related to the rate-distortion function, allowing for the solving of rate-distortion functions through their respective optimal transport solvers rather than directly.

**Comments And Feedback To The Authors:**

Good evaluation by both reviewers! Please carefully revise and proofread the paper following both reviewers' comments.

**Reason For Not Giving A Higher Recommendation:**

* Clarity of this paper could be improved.

**Reason For Not Giving A Lower Recommendation:**

* Both reviewers acknowledge the CCR and contribution of this paper.

---

### Decision · Program_Chairs · 2023-04-09

Invite to present